# Utility of Single Items within the Suicidal Behaviors Questionnaire-Revised (SBQ-R): A Bayesian Network Approach and Relative Importance Analysis

**DOI:** 10.3390/bs14050410

**Published:** 2024-05-14

**Authors:** Jenny Mei Yiu Huen, Augustine Osman, Bob Lew, Paul Siu Fai Yip

**Affiliations:** 1Department of Social Work and Social Administration, The University of Hong Kong, Hong Kong, China; h0342392@connect.hku.hk (J.M.Y.H.);; 2The Hong Kong Jockey Club Centre for Suicide Research and Prevention, The University of Hong Kong, Hong Kong, China; 3Department of Psychology, The University of Texas at San Antonio, San Antonio, TX 78249, USA; 4School of Applied Psychology, Griffith University, Mount Gravatt, QLD 4122, Australia

**Keywords:** Suicidal Behaviors Questionnaire-Revised, suicide-related thoughts and behaviors, Bayesian network modeling, relative importance analysis

## Abstract

The Suicidal Behaviors Questionnaire-Revised (SBQ-R) comprises four content-specific items widely used to assess the history of suicide-related thoughts, plans or attempts, frequency of suicidal ideation, communication of intent to die by suicide and self-reported likelihood of a suicide attempt. Each item focuses on a specific parameter of the suicide-related thoughts and behaviors construct. Past research has primarily focused on the total score. This study used Bayesian network modeling and relative importance analyses on SBQ-R data from 1160 U.S. and 1141 Chinese undergraduate students. The Bayesian network analysis results showed that Item 1 is suitable for identifying other parameters of the suicide-related thoughts and behaviors construct. The results of the relative importance analysis further highlighted the relevancy of each SBQ-R item score when examining evidence for suicide-related thoughts and behaviors. These findings provided empirical support for using the SBQ-R item scores to understand the performances of different suicide-related behavior parameters. Further, they demonstrated the potential value of examining individual item-level responses to offer clinically meaningful insights. To conclude, the SBQ-R allows for the evaluation of each critical suicide-related thought and behavior parameter and the overall suicide risk.

## 1. Introduction

About 20 years ago, the Suicidal Behaviors Questionnaire-Revised (SBQ-R) [1] was introduced to the mental health field for assessing suicide-related thoughts and behaviors. The SBQ-R consists of 171 words, is relatively short and has been found to have robust psychometric properties in clinical and non-clinical samples across cultures and nations. As a self-report instrument, the SBQ-R has the advantages of brevity, ease of administration, cost-effectiveness and objective scoring. Given the substantive contributions of the SBQ-R in the extant suicide literature, we wanted to provide additional empirical evidence for the psychometric properties of this instrument. First, we review the background information on instrument construction and the key findings from past research with the SBQ-R. Second, we discuss the ongoing concerns regarding the use of scores of Item 1 of the SBQ-R to measure or screen for suicide risk (i.e., single-item measurement). Third, we use two contemporary modeling strategies (Bayesian network modeling and relative importance analysis) to provide detailed psychometric information about the individual items of the SBQ-R.

### 1.1. Background Information on the Construction and Validation of the SBQ-R

Osman and colleagues [1] developed the SBQ-R partly because of the lack of a short-form version of the Suicidal Behaviors Questionnaire (SBQ) [2] that could be used in clinical and research settings. Furthermore, original investigations with the SBQ did not report on other important psychometric properties of the instrument, including how scores can inform clinical judgment. Therefore, Osman and colleagues examined empirical support for four clinically relevant items from the SBQ. The researchers modified the response format for each item to adapt the instrument as a brief self-report of four specific parameters of the suicide-related thoughts and behaviors construct. More specifically, Item 1 evaluates the history of suicidal thoughts, plans or attempts. The item scores range from 1 (never) to 4 (I have attempted to kill myself…to die). Item 2 assesses the frequency of suicidal ideation in the past year. The item scores range from 1 (never) to 5 (very often). Item 3 examines the communication of intent to die by suicide. The item scores range from 1 (No) to 3 (Yes, more than…to die). Item 4 estimates the self-reported likelihood of death by suicide. The item scores range from 0 (never) to 6 (very likely). Scores for all the items are summed to obtain a total scale score that ranges from 3 to 18. A higher SBQ-R total score represents a greater severity of suicide-related thoughts and behaviors.

In addition to its use as an outcome or concurrent measure, the SBQ-R was designed to screen for suicide-related thoughts and behaviors. Accordingly, the researchers proposed cutoff scores for the SBQ-R total score and Item 1 (history of suicidal thoughts and attempts). More explicitly, they used receiver operating characteristic analysis (ROC) to recommend specific cutoff scores for use with clinical and non-clinical samples for further evaluation. For example, a cutoff score of seven or higher on the SBQ-R total score was proposed as appropriate for use with nonclinical samples (Sensitivity = 93%; Specificity = 95%); however, a cutoff score of eight or higher was proposed to be adopted for use with clinical samples, including adolescents and adults (Sensitivity = 0.80% to 0.87%; Specificity = 0.91% to 0.93%). For Item 1, a cutoff score of two or higher was suggested for use with clinical and non-clinical samples (Sensitivity = 0.80% to 100%; Specificity = 0.96% to 100%).

In recent years, several systematic reviews and scholarly publications have reported on the clinical utility and psychometric properties of the SBQ-R in research and clinical settings. More noticeably, scores on the SBQ-R have contributed prominently to assessing suicide-related thoughts and behaviors. Most studies that included the SBQ-R as a concurrent or outcome measure used the SBQ-R total score as a measure of suicide risk in psychological, psychosocial or medical disorders [3,4]. Additionally, the SBQ-R has been translated into different languages and used for research in other countries. Recently, Huen et al. [5] identified a list of 57 studies that had translated the SBQ-R into 24 different languages (e.g., Arabic, Bahasa, Malay, Bangla, Burmese, Chinese, French, German, Hebrew, Italian, Japanese, Khmer, Korean, Portuguese, Russian, Spanish). Upon review, the researchers noted that most of these studies had only examined the internal consistency reliability estimates of the translated versions of the instrument. In response to the absence of research with contemporary data analytical methods, the researchers used several classical and modern data analytical strategies (e.g., multiple-group confirmatory factor analysis) to provide strong evidence for measurement invariance across cultures and the recommended cutoff score of seven for the SBQ-R total score based on cross-cultural samples of U.S. and Chinese samples. However, as in previous studies with the SBQ-R, more attention should have been given to evaluating (a) the relative importance of the individual SBQ-R item scores or (b) the relationships among scores on the individual SBQ-R items.

In a systematic review and evaluation of 19 self-report measures of suicide-related thoughts and behaviors [6], the SBQ-R was ranked among the top three self-report instruments that best met the evaluation criteria. The review focused on six commonly used psychometric criteria: internal consistency reliability, test–retest reliability, construct validity, concurrent validity, sensitivity to change and whether the instrument is free to use. However, although this objective and comprehensive review supported the psychometric properties of the SBQ-R, it did not focus on other critical psychometric properties of the instrument, such as the relative importance of the scores of the individual items within the instrument. Thus, our study aims to ensure that future literature reviews examine the relative importance of scores of the individual items within multi-item scales.

### 1.2. A Research Gap in Using the SBQ-R as a Single-Item Measurement

Recent studies (i.e., non-clinical samples) have highlighted the importance of undertaking item-level data analyses with the SBQ-R [7,8,9]. In addition to the total scores of the SBQ-R, Becker and colleagues [7] included responses from 1704 U.S. undergraduate students to the individual items of the SBQ-R to gain descriptive information about the rate of responses to each item. Lew and colleagues [8] adopted similar analyses to examine scores on the individual and total scores of the SBQ-R. Using data from undergraduate students in the U.S. (*N* = 1185) and China (*N* = 2000), the researchers examined differences in the frequency of responses to the SBQ-R individual and total scale scores. In a sample of 320 U.S. and 298 Chinese undergraduate students, Huen and colleagues [9] applied Item Response Theory modeling strategies to examine the response parameters and potential differential item functioning of the individual items of the SBQ-R. Findings from these recent studies have highlighted the value of undertaking item-level analytic strategies to explore clinically helpful information about scores of individual SBQ-R items. To this end, we decided to use modern psychometric modeling methods to evaluate the performance of scores on individual items and the total scale.

While multi-item scales are often preferable to capture the complexity of psychological constructs, single items can provide unique and clinically useful information when used separately. Single items within a multi-item scale may tap distinctive aspects of a broader construct that are not fully captured by the total score. This is particularly applicable for the SBQ-R, as each item focuses on a different parameter of suicide-related thoughts and behaviors (e.g., history, ideation, plans, likelihood). Even if the overall scale is most appropriate for assessing general risk, an individual item score could offer insight into a specific suicide risk. Additionally, brief single items may be preferable in clinical screening applications where time is limited. While a total scale score synthesizes information, the response to a single item provides a clear snapshot of an individual’s status on an essential aspect of a complex construct. This could help prioritize patients needing immediate intervention or guide treatment planning. Of note, single items are more informative when considered alongside other scale items and clinical history. With these considerations in mind, further examination of individual SBQ-R item scores is warranted.

As noted earlier, there are ongoing concerns regarding using scores on Item 1 of the SBQ-R to measure or screen for suicide risk. Specifically, it has been argued that responses to Item 1 might not be adequate for assessing the history of suicidal thoughts and attempts [10,11]. Some researchers have questioned if the item might be double-barreled [4]; others have recommended revising the question or its response options into two clusters [12,13]. Additionally, O’Dwyer et al. [14] opined that Item 1 of the SBQ-R could not connect lifetime attempts with recent life experiences. Further, Glaesmer et al. [15] noted that the differential scaling of individual items of the SBQ-R may be one of the measurement limitations of the SBQ-R. The present study attempted to address these concerns and extend empirical support for the psychometric properties of the individual SBQ-R items, particularly Item 1.

In the original paper, Osman and colleagues [1] described Item 1 of the SBQ-R as the critical item for evaluating the relative importance of the other items in the instrument. However, despite the robust findings of reliability and validity that have been reported for obtained instrument scores, none, as in the current study, evaluated other clinically valuable properties, such as the relative importance of scores for each item, as recommended by Osman et al. [1]. We reasoned further that contemporary modeling strategies would provide research and clinically useful information about the psychometric properties of the individual SBQ-R items. For example, Bayesian network modeling would inform the extent to which Item 1 plays a central or essential role for the other SBQ-R items. As another example, relative importance analysis within linear modeling would help identify the best predictors for the Item 1 scores. More explicitly, the modeling would help determine each item’s critical role relative to the Item 1 scores (established within ROC) when using this instrument. It is essential to state that prior research and reviews have yet to report on these basic psychometric properties of the SBQ-R items. Accordingly, we present brief overviews of the primary data-analytic methods to enhance understanding of each method for item-level data analyses with the SBQ-R.

### 1.3. Brief Overview of Two Contemporary Modeling Strategies (Bayesian Network Modeling and Relative Importance Analysis)

The Bayesian network uses Bayes’ theorem (conditional probability theorem) to define a probabilistic model on a random set of variables (also known as nodes) and the dependence relationships between them [16]. The modeling uses a directed acyclic graph (DAG) to represent the model network structure. For example, the DAG in Figure 1 shows the dependence relationships linking four variables of the SBQ-R, labeled after the item numbers (SBQ1, SBQ2, SBQ3 and SBQ4). Direct dependency between two nodes, such as SBQ-R Item 4, depends on SBQ-R Item 1, represented by an arc pointing in one direction from one node to another (i.e., SBQ1 → SBQ4). The node at the beginning of the arc (i.e., where the arrow points from) is known as the parent node (e.g., SBQ-R Item 1 in this example); the node at the end of the arc (i.e., where the arrow points to) is known as a child node (e.g., SBQ-R Item 4 in this example). More precisely, DAG is acyclic because its arcs are not allowed to have any directed cycle, starting and ending at the same node after a sequence of arcs (e.g., SBQ1 → … → SBQ1).

Based on the probability concept, Bayesian network modeling has an assumption or property known as the Markov property, stating that any dependence relationships between nodes must take the form of arcs. Accordingly, two nodes are linked by an arc if one is conditionally dependent on the other; otherwise, they would be conditionally independent because of the lack of an arc between them. A node with arcs pointing to other nodes without an arrow pointing to it from other nodes can be identified as the cause of the problem being studied (known as the root node—a node with children but without parents). Root nodes have marginal distributions, which are their probability distributions defining the probabilities of all their possible values without considering the values of other nodes. In contrast, a node with arcs pointing to them from other nodes but without arcs pointing to other nodes can be identified as the final effect of the problem being studied (known as a leaf node—a node with parents but without children).

The probability distributions of leaf nodes are conditional on the values of their respective parents. While the arcs inform the probabilistic dependencies between nodes, the dependencies’ strengths are quantified by conditional probability distributions for discrete variables. Using the same example, the conditional probability distribution of SBQ-R Item 4 for SBQ-R Item 1 is its probability distribution when Item 1 equals its discrete values. The discrete value can be 1, 2, 3 or 4 because the Item 1 scores range from 1 to 4. Thus, the conditional probability distribution of Item 4 depends on the value of Item 1; knowing the value of Item 1 modifies our knowledge about the distribution of Item 4. In particular, the probability that Item 4 will have each discrete value can be calculated by looking at the possible joint values that Item 1 has.

Bayesian networks can be undertaken for various specific aims and purposes, including predictive, diagnostic and descriptive. Indeed, it can even be used as an exploratory strategy to study symptoms [17,18]. The application of Bayesian networks in psychopathology research has recently been recommended by Briganti et al. [19] to gain insight into the dependent or admissible causal relationships among symptoms. Suicide-related thoughts and behaviors, such as suicidal ideation, suicide plans and suicide attempts, are likely to result from a complex interaction of multiple factors [20,21,22]. Understanding the dependence relationships between these constructs is essential to inform diagnosis and prediction. The Bayesian network makes it possible to make probabilistic statements about a specific construct (e.g., a lifetime of suicide thoughts, plans and attempts) through the DAG or the corresponding probability distributions. 

Bayesian techniques like those employed in this study can help address the reproducibility crisis in the field. As noted by Etz and Vandekerckhove [23], conventional frequentist statistics inflate prediction performance and do not advance suicide-related investigations. Bayesian techniques overcome these limitations through their flexibility and focus on model evaluation over hypothesis testing alone. Nguyen et al.’s study [24] exemplifies how Bayesian techniques can improve reproducibility in suicide-related research. They employed Bayesian techniques to analyze data collected using a single item from the PHQ-9, which assesses suicidal thoughts over the past two weeks and explore suicidal ideation mechanisms by evaluating the associations with a sense of connectedness and help-seeking behaviors in four different models. In their study, Bayesian techniques provided advantages like flexibility, emphasis on model evaluation over hypothesis testing and enhanced capacity to reproduce findings in psychology research. We note further that machine learning algorithms can help address some of the methodological limitations of conventional statistical methods in studying the suicide-related thoughts and behaviors construct [20,22,25]. In particular, Bayesian network modeling allows for simultaneously examining several variables and probabilistic relationships via learning algorithms. The process or analysis considers all possible links between the variables and identifies an optimal model that maximizes the accuracy in modeling the variables’ associations. In contrast, conventional approaches such as confirmatory factor analysis and structural equation modeling methods extended to the SBQ-R items only allow for testing the structural relationships between hypothesized variables; they do not inform all the other possible or more complex relationships present in item-level data.

The relative importance or contribution percentage for each predictor to the coefficient of determination (R^2^; the proportion of variance explained by the whole set of predictors) in a multiple linear regression model can be estimated by averaging the R^2^ contributed across all possible combinations of the orderings of predictors [26]. Also, a bootstrapping procedure can be used within the relative importance modeling to compare the predictors’ scores. Together, relative importance analysis with bootstrapping can identify the best set of significant predictors useful for the regression model.

Using the relative importance analysis within regression modeling can also increase the prediction accuracy and parsimony of the model as an alternative to conventional regression analysis. Moreover, it reduces response biases associated with the item or scale scores [27,28,29]. The reasons, in part, are as follows. First, the sequence in which predictors are entered into the model can affect the *R*^2^ contributed by the individual predictors. Second, the regression coefficients, including standardized regression coefficients, cannot be directly compared, given the existence of multicollinearity and response bias. Third, measurement error is addressed directly in the metrics used in the relative importance modeling, which outweighs the conventional Pearson correlation or multiple regression modeling, which focuses on standardized coefficient betas but not the errors. Given an empirically supported criterion variable, the procedure informs researchers and clinicians about the individual predictors’ relative importance.

### 1.4. Overview of the Study

The SBQ-R is a widely used brief self-report measure for assessing suicide-related thoughts and behaviors. While extensive research supports the measure’s psychometric properties, most studies have focused only on the total scale score. However, examining individual item-level responses could provide clinically meaningful insights beyond a total score approach.

The current study uses contemporary modeling techniques to validate the utility of individual SBQ-R item scores. The primary goals were to (1) examine the dependence relationships among the individual SBQ-R items and (2) evaluate the importance of how each of the other items functions within the SBQ-R relative to scores on Item 1 (history of suicidal thoughts and attempts). Bayesian network analysis was undertaken to examine if the frequency of suicidal ideation (Item 2), communication of intent to die by suicide (Item 3) and self-reported likelihood of a suicide attempt (Item 4) are dependent on the history of suicidal thoughts and attempts (Item 1). In addition, each item’s relative importance as a predictor of the Item 1 scores was estimated and bootstrapped. The analysis was extended to a Chinese and a U.S. sample to ensure the results hold for a geographically diverse population. No prior research has evaluated these psychometric properties of the SBQ-R items. Results can help inform optimal use of the SBQ-R in both research and practice settings.

## 2. Materials and Methods

The characteristics of the participants are provided in Table 1. The U.S. undergraduate sample consisted of 1160 students (425 men, 725 women, 10 transgender or not sure) aged between 18 and 58 years (*M* = 20.12, *SD* = 3.43) recruited from a large Southwestern state university. The participants completed the survey online to partially fulfill the psychology course requirements. Their current years of undergraduate education are Freshman (54.8%), Sophomore (21.1%), Junior (14.5%) and Senior (9.6%). The sample self-identified as Latino/Latina (42.1%), White (30.1%), African American/Black (10.7%), Asian/Pacific Islander American (6.2%), Multiracial (5.4%) and other ethnic/racial groups (5.5%).

The Chinese undergraduate sample consisted of 1141 students randomly drawn with no replacements from a pool of 10,961 undergraduate students (about 10% of the pool). The random sampling procedure was undertaken to minimize the potential impact of the size differences between the U.S. and Chinese samples. The sample included 455 men and 686 women; they ranged in age from 18 to 27 (*M* = 20.88, *S.D.* = 1.52). The participants completed the survey voluntarily. They were enrolled in various undergraduate degrees at universities in four Chinese provinces (Jilin, Qinghai, Shaanxi and Shandong), two autonomous regions (Ningxia Hui and Xinjiang Uygur) and one municipality (Shanghai). Their current years of undergraduate education are Freshman (29.8%), Sophomore (23.7%), Junior (20.2%) and Senior (8.3%); 18.1% of them did not provide relevant information. Most of the participants self-identified as Han Chinese (74.7%). The Han Chinese constitutes the largest ethnic group in China, and non-Han Chinese are ethnic minorities.

The Southwestern State University’s Institutional Review Board (IRB) reviewed and approved the study protocol for the U.S. undergraduate sample (No. 17-206N). The survey (containing questions about demographics and study self-report instruments) was set up in the Qualtrics survey platform. Participants had to log into the Qualtrics survey platform and provide informed consent before taking the survey. The system prompts the participant to answer all the questions if a question is not endorsed. Also, the system would prevent multiple submissions from the same participant. As a result, there was no missing data from each participant or various entries from the same participant.

The study with the Chinese undergraduate group met the national ethical review requirements for approval. The study protocol was approved by the Ethics Review Committee of Ningxia Medical University (No. 2017-171). Data were collected from nine universities across four of the 23 provinces (Jilin, Qinghai, Shaanxi and Shandong), one of the four municipalities (Shanghai) and two of the five autonomous regions (Ningxia and Xinjiang). Within each university, approximately 50 classes were sampled. All students in the selected classes (about 40 students per class) were invited to participate in the paper survey.

Participation was voluntary and anonymous and could be withdrawn if the participant desired. The response rate was approximately 90%. Informed consent was obtained from all study participants. Information on hotline counseling services was also provided should a participant seek professional help. Data were excluded for participants who did not complete the study measures, who did not provide essential demographic information, including age and gender, or who were out of the normative age range of undergraduate students in the education system in China (i.e., 18 to 27 years). The final sample consisted of 10,961 Chinese undergraduate students. As aforementioned, a random subsample of approximately 1150 Chinese undergraduate students was drawn from the total sample (*N* = 1141; 455 men and 686 women).

In addition to the SBQ-R, a basic demographic information questionnaire, including age, sex, years of undergraduate education and ethnicity, was included in the survey with the U.S. and Chinese undergraduate groups. The original (English) version of the SBQ-R was used with the U.S. sample. The Chinese version of the SBQ-R was used with the Chinese sample (detailed information regarding the translation and back-translation steps for the Chinese version are reported [8]). The Shandong University Centre for Suicide Prevention Research translated the Chinese version of the SBQ-R, and the instrument has been validated by Huen et al. [5] for use with Chinese undergraduate students using a culture, comprehension and translation bias procedure [30]. 

Item statistics, including response frequencies, descriptive statistics and internal consistency reliability estimates of the SBQ-R scores, were computed using JASP version 0.14.3 [31]. The internal consistency reliability estimate was estimated by coefficient-omega (*ω*) [32] and coefficient-alpha (*α*) [33] using the Bayesian approach [34]. Unlike the frequentist approach, the Bayesian approach analyzes all unknown parameters as random variables in estimating internal consistency reliability. Accordingly, this approach was used to obtain a credible interval, where, to a certain extent, the actual population reliability coefficient lies (i.e., the frequentist approach bootstraps the confidence interval, which, to a certain extent, could capture the true reliability coefficient) [35]. We adopted the minimum “standard of reliability” estimate of 0.70, as Nunnally [36] proposed to indicate acceptable internal consistency reliability.

The Bayesian network was constructed with a bnlearn package [37] developed in R software version 4.0.5 [38]. The SBQ-R (Item 1 to Item 4) variables were entered as discrete data to construct the Bayesian networks for the U.S. and Chinese undergraduate groups. A hill-climbing greedy search algorithm [39] was used to learn the Bayesian network structure from the data. This score-based learning algorithm scores each potential Bayesian network associated with the data and searches for an optimal Bayesian network structure that maximizes the score. A DAG was plotted from the Bayesian network analysis to show each group’s resultant Bayesian network structure, representing a set of the four variables and their conditional dependencies (if any). A bootstrap approach was applied to estimate the strength of the conditional dependence corresponding to each arc [40], quantified by the probability of a dependence relationship between each pair of nodes [41]. For each node, the conditional probabilities of all the possible combinations of its parent nodes are listed in a conditional probability; the results are presented in a table for each group. 

The K-fold cross-validation approach [42] was applied to randomly divide each sample data into 10 subsets (i.e., 10-fold cross-validation). Each subset is then used to validate the Bayesian network model fitted on the remaining nine subsets to obtain an unbiased estimate of the model’s goodness-of-fit in terms of log-likelihood loss. A smaller log-likelihood loss indicates a better model fit. 

The relative importance of SBQ-R Items 2, 3 and 4 scores in predicting the SBQ-R Item 1 scores in a regression model was examined using the relaimpo package [43] in R [38]. The predictor scores were standardized for the analysis. A bootstrap procedure was undertaken. Specifically, the procedure allows for comparing the relative importance of the predictor scores, as measured by the *R*^2^. Moreover, the method provides plots of the bootstrap 95% confidence intervals (C.I.s) for the differences in the *R*^2^ of each predictor. 

The K-fold cross-validation was performed with the caret package [44] developed in R [38]. Using a 10-fold cross-valuation, we evaluated the model’s prediction on each of the 10 subsets using the following metrics: *R*^2^, root mean square error (RMSE), and mean absolute error (MAE). The larger the *R*^2^ and the lower the prediction errors (i.e., RMSE and MAE), the better the model’s prediction.

The above analyses were conducted separately for the U.S. and Chinese undergraduate groups. 

## 3. Results

### 3.1. Descriptive Statistics and Internal Consistency Reliability

Descriptive statistics for the SBQ-R item and total scale scores are provided in Table 1. The internal consistency reliability of the SBQ-R total scores, estimated by Bayesian coefficient-omega and Bayesian coefficient-alpha, were adequate. In the U.S. undergraduate sample (*N* = 1160): Bayesian-*ω* = 0.854 (95% CI = 0.840, 0.868) and coefficient-*α* = 0.829 (95% CI = 0.815, 0.842). In the Chinese undergraduate sample (*N* = 1141): Bayesian-*ω* = 0.780 (95% CI = 0.758, 0.801) and coefficient-*α* = 0.753 (95% CI = 0.733, 0.773).

### 3.2. Bayesian Network Analysis

The Bayesian network learned via the structure learning algorithm had the same resultant structure for the U.S. and Chinese undergraduate groups, as shown in the DAG (Figure 1 as a model network structure). There are a total of 3 directed arcs leading from the SBQ-R Item 1 to Item 2 (arc strength = 1 for both the U.S. and Chinese groups), Item 3 (arc strength = 1 for the U.S. group and 0.99 for the Chinese group) and Item 4 (arc strength = 0.97 for the U.S. group and 1 for the Chinese group), respectively. Results suggested that the frequency of suicidal ideation (Item 2), communication of intent to die by suicide (Item 3) and self-reported likelihood of a suicide attempt (Item 4) may be strongly dependent on the history of suicidal thoughts and attempts (Item 1) for each group.

The conditional probability distributions corresponding to the DAG are reported for the U.S. undergraduate group (Table 2) and the Chinese undergraduate group (Table 3). For easy reference, the conditional probability distributions are separated into a set of conditional probability tables, one for each variable, for the U.S. group (Appendix A) and the Chinese group Appendix A available online. SBQ-R Item 1 is a root node (i.e., a node with children but without parents) modeled by a unidimensional probability table computed from its empirical frequencies (see Table 1). SBQ-R Items 2, 3 and 4 are leaf nodes (i.e., nodes with parents but without children) that depend only on SBQ-R Item 1 and are modeled by their corresponding two-dimensional conditional probability tables. Each two-dimensional conditional probability table column corresponds to one level of Item 1, showing the distribution of Item 2/Item 3/Item 4 conditional on that particular level of Item 1. As a result, the probabilities sum up to 1 within each column.

Based on the conditional probability distributions (Table 2) or tables (Appendix A), the probability of suicide-related thoughts and behaviors (measured by Item 2, Item 3 and Item 4) differs conditionally on the range of history of suicidal thoughts and attempts (measured by Item 1). First, taking the arc from Item 1 to Item 2, in the U.S. group as an example, the conditional probability of having the lowest score “1 (Never)” on Item 2 (frequency of suicidal ideation) is 0.99 given the lowest score “1 (Never)” on Item 1 (history of suicidal thoughts and attempts). Thus, we can be almost sure that those who do not have a history of suicidal thoughts and attempts are not susceptible to suicidal ideation. Second, the conditional probability of having the lowest score of “1 (Never)” or second-lowest score of “2 (1 time)” on Item 2 (frequency of suicidal ideation) is about 0.80, given the second-lowest score of “2 (passing thought)” on Item 1 (history of suicidal thoughts and attempts). Accordingly, there is an increased likelihood for individuals with histories of brief passing thoughts of suicide to report a higher frequency of suicidal ideation in the past year than those with no history of suicidal thoughts and attempts. If individuals with histories of brief passing thoughts of suicide indeed have suicidal ideation in the past year, it is most likely to be low in frequency. Third, the conditional probability of having mid-range scores “2 (1 time)”, “3 (2 times)” or “4 (3–4 times)” on Item 2 (frequency of suicidal ideation) is about 0.70, given the second-highest score “3 (had a plan)” on Item 1 (history of suicidal thoughts and attempts). Thus, there is an increased likelihood for individuals with histories of suicide plans to have suicidal ideation in higher frequency than those with only a history of brief thoughts of suicide. Finally, the conditional probability of having the higher scores “3 (2 times)”, “4 (3–4 times)” or “5 (5 or more times)” on Item 2 (frequency of suicidal ideation) is about 0.60, given the highest score “4 (attempted)” on Item 1 (history of suicidal thoughts and attempts). Accordingly, there is a higher likelihood for individuals with histories of suicide attempts to have a moderate to high frequency of suicidal ideation.

As another example, the Chinese group’s arc from Item 1 to Item 3 is used to interpret the corresponding probabilities from the conditional probability distributions (Table 3) or tables (Appendix A). The conditional probability of having the lowest score of “1 (No)” on Item 3 is 0.98, given the lowest score of “1 (Never)” on Item 1 (history of suicidal thoughts and attempts). From this finding, we can be almost sure that those who do not have a history of suicidal thoughts and attempts are unlikely to make a suicide threat. In contrast, the conditional probabilities of having the highest and second-highest scores, “2 (at one time)” or “3 (more than once)” on Item 3 (communication of intent to die by suicide), are about 0.80, given the highest score, “4” on Item 1 (history of suicidal thoughts and attempts). This analysis shows an increased likelihood for individuals with histories of suicide attempts to make one or more suicide threat(s).

Similar patterns of conditional probabilities can be observed for the other pairs of nodes in both the U.S. and Chinese groups. The 10-fold cross-validation for this Bayesian network showed that the expected loss in log-likelihood was 3.31 and 2.54 for the U.S. and Chinese groups, respectively. These findings indicate that the network model had an excellent overall fit to the dependence structure of the data.

The Bayesian network analysis provides further evidence that individual SBQ-R item scores can offer unique clinical value when examined alongside the other items. Specifically, Item 1 appears to serve as an essential foundational item tapping into the history of suicidal thoughts and behaviors. This single item alone activates responses on the other items regarding ideation, communication and likelihood. Considering Item 1 responses during clinical screening could help group patients according to the level of prior risk experienced and guide the direction of further assessment and treatment planning.

### 3.3. Relative Importance Analysis within Regression

Relative importance analysis within multiple linear regression was conducted to determine the relative importance of SBQ-R Items 2, 3 and 4 scores to predict scores on Item 1. About 50% to 60% of the variance (i.e., 56.21% for the U.S. undergraduate group and 51.07% for the Chinese undergraduate group) could be accounted for in the multiple regression model with Item 2 (frequency of suicidal ideation), Item 3 (communication of intent to die by suicide) and Item 4 (self-reported likelihood of a suicide attempt) scores predicting Item 1 (history of suicidal thoughts and attempts) scores of the SBQ-R. Among the three predictors, Item 2 contributed 20.76%, Item 3 15.77% and Item 4 19.68% for the U.S. group, whereas for the Chinese group, Item 2 contributed 20.25%, Item 3 15.48% and Item 4 15.34%. The relative importance values of Item 2, Item 3 and Item 4 were 0.37, 0.28 and 0.35, respectively, for the U.S. group. For the Chinese group, these values were 0.39 (Item 2), 0.30 (Item 3) and 0.30 (Item 4).

The relative importance values with bootstrap C.I.s for the U.S. and Chinese groups are plotted in Figure 2. In each group, a predictor’s lower- and upper-C.I. bounds cover the *R*^2^ of the other predictors. Accordingly, there is no difference in relative importance for each pair of predictors. For example, for the Chinese group, the lower- and upper-CI’s for Item 2’s contribution to the overall *R*^2^ (0.20) is 0.14 to 0.27, covering the *R*^2^ for Item 3 and Item 4 (both 0.15). Similarly, because all the C.I.s for the differences between the relative contributions cover 0, the differences of each pair of predictors are not statistically significant in both groups. For the Chinese group, the difference in *R*^2^ between Item 2 and Item 3 was 0.05 [−0.02, 0.12], between Item 2 and Item 4 was 0.05 [−0.02, 0.12] and between Item 3 and Item 4 was 0.01 [−0.07, 0.06]. For the U.S. group, the difference in *R*^2^ between Item 2 and Item 3 was 0.05 [−0.02, 0.11], between Item 2 and Item 4 was 0.01 [−0.05, 0.07] and between Item 3 and Item 4 was −0.04 [−0.12, 0.04].

The *R*^2^ and prediction errors for the regression model, estimated with the 10-fold cross-validation method, were *R*^2^ = 0.51, RMSE = 0.49 and MAE = 0.33 for the Chinese group and *R*^2^ = 0.51, RMSE = 0.65 and MAE = 0.49 for the U.S. group. The relative importance analyses consistently showed that no predictor is significantly more important than the other; all three SBQ-R item scores (Items 2, 3 and 4) were important in predicting the Item 1 scores.

The relative importance analysis indicates that all three other SBQ-R items remain important predictors of Item 1 scores. This suggests that each item provides at least some independent information regarding suicidal history beyond what is conveyed in the other items and the total scale score. While the full scale should still be preferred for comprehensive assessment, these findings support further exploration of how individual item scores might complement total scores to enhance the clinical and research utility of the SBQ-R.

## 4. Discussion

### 4.1. Key Findings of This Study

Despite current work (reviews and studies) supporting the psychometric properties of the SBQ-R, the importance of scores on the individual items has not been examined adequately. In this study, we adopted alternative modeling methods to validate the performances of SBQ-R items’ scores. In addition to commonly reported internal consistency reliability estimates, we examined the dependence relationships between the individual item parameters. First, we conducted the Bayesian network analysis to evaluate the performance of the SBQ-R Item 1 responses relative to scores on the other items (i.e., Items 2, 3 and 4). When there are multiple risk factors (observed indicators: ideation, threat, self-reported likelihood) that are linked with a single outcome (i.e., history of suicidal thoughts and attempts), Bayesian network modeling, as undertaken in this study, estimates conditional probabilities to examine the direction and the magnitude of dependence relationships of these risk factors with the history of suicidal thoughts and attempts (Item 1). Results indicated that in both the U.S. and Chinese undergraduate groups, the frequency of suicidal ideation (Item 2), communication of intent to die by suicide (Item 3) and self-reported likelihood of a suicide attempt (Item 4) are dependent on responses to the history of suicidal thoughts and attempts (Item 1) in large magnitude. In addition, Items 2 to 4 are conditionally independent given Item 1 (i.e., after controlling for Item 1). Accordingly, a history of suicide-related thoughts and behaviors can be regarded as a potential common source of activation for other parameters of the suicide-related thoughts and behaviors construct, as assessed with the SBQ-R.

For future investigations, using the DAG (as in Figure 1) and the conditional probability distributions (as in Table 2 and Table 3) in Bayesian network modeling might offer additional perspectives for using the SBQ-R in research and clinical settings. For example, based on a piece of intake information about the history of suicidal behavior (Item 1), it is possible to formulate meaningful hypotheses about the other parameters of the suicide-related thoughts and behaviors constructed on the network structure. The probabilistic inference supports the original recommendation that the SBQ-R Item 1 can be used as a screening item to form subgroups of study participants [1], despite some concerns previously raised regarding using this single item. More importantly, it might inform clinical judgment about the relationships between each SBQ-R item (e.g., frequency of ideation and likelihood of attempts) and the history of suicidal behaviors (Item 1).

It is noteworthy that our results are consistent with most of the findings in the extant literature. For example, the identification of SBQ-R Item 1 as the source of activation, which maintains other more recent suicide-related thoughts and behaviors, supports previous conclusions that past suicide attempts remain one of the best predictors or indicators of future suicide attempts in both Western and Chinese contexts [22,45,46,47]. Chen and his colleagues [45] found in the first territory-wide psycho-autopsy study in the Chinese context that having past suicide attempts vastly increased one’s risk of suicide (odds ratio about 25). In the Western context, Suokas and colleagues [47] followed up with 1018 patients after 14 years. They found from the survival analysis results that past suicide attempts continued to be a significant risk factor for suicide. Accordingly, individuals who have attempted suicide in the past could be considered potential targets and given higher priority for clinical interventions.

One prominent finding or advantage of using a single SBQ-R item lies in its capability to be analytically studied and incorporated with other theories and frameworks, as demonstrated in Nguyen et al.’s study [24], where a single PHQ-9 item measuring suicidal ideation fits logically within the Mindsponge theoretical framework [48]. The Mindsponge theory [48] conceptualizes cognition as an evolving information filtering process, aligning well with operationalizing one element (e.g., suicidal thoughts) via a single-item Likert scale. This compatibility permitted logically justifying associations between variables like a sense of belongingness influencing suicidal ideation through help-seeking behaviors. Similarly, Item 1 of the SBQ-R assesses the critical history of suicidal thoughts, plans and attempts parameter. It can be more parsimoniously analyzed and modeled within a theoretical framework as a single-item measure compared to total scale scores. Specifically, the Bayesian network showed that Item 1 served as a root node tapping into past suicidal risk, allowing formulations of how other ideation and behavioral parameters may be conditionally influenced. Focusing analytical attention on a single item rather than the full scale could give deeper insight into the role and relationships of the unique parameter assessed by Item 1. This enhances our understanding of how the SBQ-R may be optimally applied and incorporated with other theories and frameworks.

Further, our findings highlight the importance of undertaking replication studies across independent modeling methods to enhance understanding of using the SBQ-R effectively in a non-clinical setting, especially where the base rate for suicidal behaviors can be low. Results from the relative importance analysis (with the bootstrap procedure) within a multiple linear regression model showed that scores on each SBQ-R item (i.e., Items 2, 3 or 4) could be a viable predictor of Item 1 responses. Thus, these three items were also *necessary* or valuable within the Bayesian network and the relative importance methods.

One of the significant measurement limitations of the SBQ-R that has not been addressed fully relates to the value of the differential response options of the SBQ-R items. Using the Bayesian network, we modeled the SBQ-R items at their differential scores (scaling) of suicide-related thoughts and behaviors. This study’s data analytic procedures and findings offered alternative ways of exploring the characteristics of the SBQ-R and evaluating the performances of the suicide-related thought and behavior parameter measured by the SBQ-R. This approach moves beyond using or validating the SBQ-R total scores to assess suicide risk.

Furthermore, the Bayesian data-analytic procedure used in this study could be used to study specific types of psychopathologies. Researchers could find the procedure helpful for (1) handling differential scaling issues within an instrument intended to assess important (unique) aspects of the construct of interest, (2) addressing how the parameters are expressed/linked as multiple processes by considering probabilistic relationships between the parameters and (3) identifying both the direction and the magnitude of dependent relationships among different variables in cross-sectional data. Briganti et al. [19] recently provided a guided tutorial modeling Bayesian networks with empirical data and relevant codes. Indeed, Bayesian network modeling may be a viable machine learning model for consideration, given that Jacobucci et al. [49] demonstrated that decision trees/ random forests within machine learning models had inflated prediction performance and might not advance suicide-related investigations.

### 4.2. Study Limitations and Future Directions

Finally, the study’s following limitations would need to be considered when researching the SBQ-R in the future. First, the current findings may not generalize to clinical or non-clinical groups other than the young adult study participants in the U.S. and China in this research. Because non-clinical samples, ages 18 and older, present with a range of at-risk behaviors, the study’s findings might contribute to conceptualizing primary prevention strategies for this age group. To go beyond the heuristic value of the clinical application of the SBQ-R provided in this study, future studies could be undertaken to replicate this research in clinical samples. 

Second, this research was based on cross-sectional data collected via self-report measures. Though suicide ideation, plan, threat and attempts are constructs that do not lend themselves readily to longitudinal investigations, future meta-analyses could synthesize findings across longitudinal studies. Cross-sectional studies can only collect responses from individuals who make a non-fatal suicide attempt. Psycho-autopsy studies are needed to capture the characteristics of individuals who make fatal suicide attempts. 

Third, the research focused on only four specific parameters of suicide-related thoughts and behaviors, as measured by the SBQ-R. This research did not examine the performance of the SBQ-R total scores and other suicide-related thought and behavior parameters such as intent and self-harm. However, they could be analyzed using similar procedures with responses on new or existing instruments designed to measure these parameters in future studies. Similarly, a meta-analytic approach could be used to establish the relationship between various risk factors with suicide-related thoughts and behaviors [50]. Also, the link between other relevant risk and protective factors and other theories of suicide could be explored using the analytic approach involving Bayesian network modeling. For future direction, suicide-related investigations could use Bayesian network modeling to identify clusters of risk factors (e.g., history of attempts and current ideation) and protective factors (e.g., social belongingness and reasons for living) to establish nomological network models as potential clusters of predictors of the target construct (e.g., suicide).

### 4.3. Conclusions

The current study enhances our understanding of how the SBQ-R may be best utilized in research and clinical practice. In addition to confirming the importance of the total scale score, results demonstrate the potential value of examining responses at the individual item level. As discussed, unique aspects of suicidal risk tapped by single items should not be overlooked, especially when scales assess complex constructs like suicide. The Bayesian network and relative importance analyses in this study help further validate how specific SBQ-R items, like the history of attempts, independently characterize aspects of risk beyond the full scale. With appropriate use, individual item scores may offer clinically meaningful insights to complement a total score approach.

## Figures and Tables

**Figure 1 behavsci-14-00410-f001:**
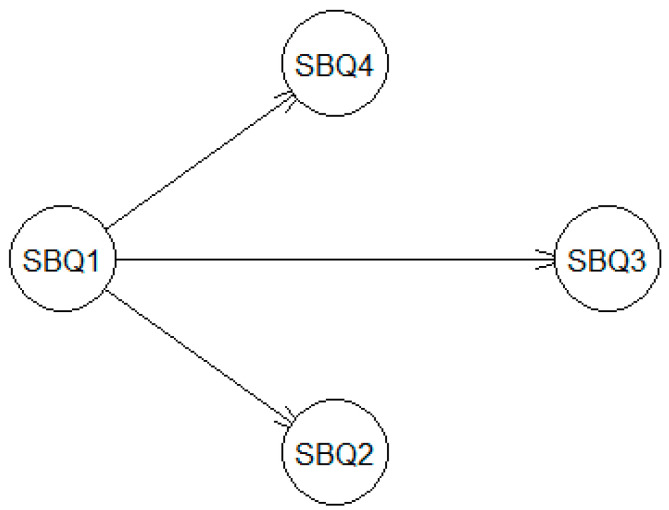
A directed acyclic graph showing the dependence relationships linking the variables of the SBQ-R. SBQ1 = history of suicidal thoughts and attempts), SBQ2 = frequency of suicidal ideation, SBQ3 = communication of intent to die by suicide and SBQ4 = self-reported likelihood of a suicide attempt.

**Figure 2 behavsci-14-00410-f002:**
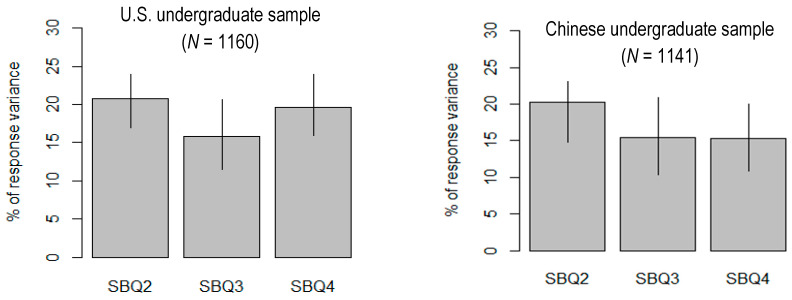
Plots of the relative importance values with bootstrapping C.I.s in predicting SBQ-R Item 1 (history of suicidal thoughts and attempts). C.I.s = confidence intervals; SBQ2 = SBQ-R Item 2 (frequency of suicidal ideation); SBQ3 = SBQ-R Item 3 (communication of intent to die by suicide); SBQ4 = SBQ-R Item 4 (self-reported likelihood of a suicide attempt).

**Table 1 behavsci-14-00410-t001:** Characteristics of the samples in the current study.

Variables	U.S. Sample (*N* = 1160)		Chinese Sample (*N* = 1141)	
	*M* or *n*	*S.D.* or %	*M* or *n*	*S.D.* or %
Age	20.12	3.43	20.88	1.52
Gender				
Males	425	36.6%	455	39.9%
Females	725	62.5%	686	60.1%
Transgender/ Not sure	10	0.9%	-	-
Year of Undergraduate Education				
Freshman (Year 1)	636	54.8%	340	29.8%
Sophomore (Year 2)	245	21.1%	270	23.7%
Junior (Year 3)	168	14.5%	230	20.2%
Senior (Year 4 or above)	111	9.6%	95	8.3%
Not reported	-	-	206	18.1%
Ethnicity				
African American/ Black	124	10.7%	-	-
Asian/ Pacific Islander American	72	6.2%	-	-
Han Chinese	-	-	852	74.7%
Latino/Latina	488	42.1%	-	-
White/Caucasian	349	30.1%	-	-
Multiracial	63	5.4%	-	-
Non-Han Chinese	-	-	278	24.3%
Others/Not reported	64	5.5%	11	1.0%
Suicide-Related Behaviors (as measured by the SBQ-R scores)	5.47	3.08	4.49	2.39
History of suicidal thoughts and attempts (as measured by the SBQ-R Item 1 scores)	1.81	0.93	1.41	0.70
Never (Score = 1)	559	48.2%	793	69.5%
Passing thought (Score = 2)	339	29.2%	253	22.2%
Had a plan (Score = 3)	185	15.9%	73	6.4%
Attempted (Score = 4)	77	6.6%	22	1.9%
Frequency of suicidal ideation (as measured by the SBQ-R Item 2 scores)	1.68	1.09	1.27	0.66
Never (Score = 1)	738	63.6%	928	81.3%
1 time (Score = 2)	199	17.2%	145	12.7%
2 times (Score = 3)	121	10.4%	41	3.6%
3–4 times (Score = 4)	58	5.0%	22	1.9%
5 or more times (Score = 5)	44	3.8%	5	0.4%
Communication of intent to die by suicide (as measured by the SBQ-R Item 3 scores)	1.28	0.56	1.15	0.42
No (Score = 1)	897	77.3%	994	87.1%
At one time (Score = 2)	200	17.2%	122	10.7%
More than once (Score = 3)	63	5.4%	25	2.2%
Self-reported likelihood of a suicide attempt (as measured by the SBQ-R Item 4 scores)	0.70	1.11	0.66	1.18
Never (Score = 0)	743	64.1%	768	67.3%
No chance at all (Score = 1)	169	14.6%	187	16.4%
Rather unlikely (Score = 2)	158	13.6%	67	5.9%
Unlikely (Score = 3)	46	4.0%	54	4.7%
Likely (Score = 4)	37	3.2%	56	4.9%
Rather likely (Score = 5)	6	0.5%	7	0.6%
Very likely (Score = 6)	1	0.1%	2	0.2%

**Table 2 behavsci-14-00410-t002:** Bayesian network analysis on the SBQ-R item scores: Conditional probability distributions for the U.S. undergraduate sample (*N* = 1160).

	SBQ1 = 1	SBQ1 = 2	SBQ1 = 3	SBQ1 = 4
SBQ2 = 1	0.99	0.40	0.20	0.19
SBQ2 = 2	0.01	0.38	0.27	0.19
SBQ2 = 3	0	0.16	0.26	0.21
SBQ2 = 4	0	0.04	0.16	0.18
SBQ2 = 5	0	0.02	0.11	0.22
SBQ3 = 1	0.98	0.73	0.44	0.29
SBQ3 = 2	0.02	0.25	0.43	0.35
SBQ3 = 3	0.01	0.02	0.13	0.36
SBQ4 = 0	0.89	0.58	0.19	0.19
SBQ4 = 1	0.08	0.24	0.16	0.13
SBQ4 = 2	0.03	0.15	0.39	0.23
SBQ4 = 3	0	0.02	0.15	0.13
SBQ4 = 4	0	0	0.08	0.27
SBQ4 = 5	0	0	0.02	0.03
SBQ4 = 6	0	0	0	0.01

Note. SBQ1 = SBQ-R Item 1 (history of suicidal thoughts and attempts); SBQ2 = SBQ-R Item 2. frequency of suicidal ideation); SBQ3 = SBQ-R Item 3 (communication of intent to die by suicide); SBQ4 = SBQ-R Item 4 (self-reported likelihood of a suicide attempt).

**Table 3 behavsci-14-00410-t003:** Bayesian network analysis on the SBQ-R item scores: Conditional probability distributions for the Chinese undergraduate sample (*N* = 1141).

	SBQ1 = 1	SBQ1 = 2	SBQ1 = 3	SBQ1 = 4
SBQ2 = 1	0.97	0.51	0.29	0.32
SBQ2 = 2	0.02	0.40	0.32	0.18
SBQ2 = 3	0	0.06	0.25	0.23
SBQ2 = 4	0	0.04	0.12	0.14
SBQ2 = 5	0	0	0.03	0.14
SBQ3 = 1	0.98	0.72	0.41	0.23
SBQ3 = 2	0.02	0.26	0.47	0.45
SBQ3 = 3	0	0.02	0.12	0.32
SBQ4 = 0	0.79	0.49	0.16	0.23
SBQ4 = 1	0.15	0.20	0.19	0
SBQ4 = 2	0.03	0.15	0.11	0.09
SBQ4 = 3	0.02	0.08	0.21	0.09
SBQ4 = 4	0.01	0.08	0.26	0.45
SBQ4 = 5	0	0	0.07	0.09
SBQ4 = 6	0	0	0	0.05

Note. SBQ1 = SBQ-R Item 1 (history of suicidal thoughts and attempts); SBQ2 = SBQ-R Item 2 (frequency of suicidal ideation); SBQ3 = SBQ-R Item 3 (communication of intent to die by suicide); SBQ4 = SBQ-R Item 4 (self-reported likelihood of a suicide attempt).

## Data Availability

The datasets generated during and/or analyzed during the current study are not publicly available due to some participants’ refusal for their data to be shared publicly.

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
