# Peer review of "Utility of Single Items within the Suicidal Behaviors Questionnaire-Revised (SBQ-R): A Bayesian Network Approach and Relative Importance Analysis"

_behavsci, 2024, doi:10.3390/bs14050410_

Round 1

Reviewer 1 Report

Comments and Suggestions for Authors

The study conducted the Bayesian network analysis and relative importance analysis on the US and Chinese datasets to indicate the value of each single item in the Suicidal Behaviors Questionnaire-Revised (SBQ-R). Some major problems need to be addressed.

- First, the rationale and significance of the study are ambiguous. The authors need to indicate more value of single-item usage.

- Second, one merit of the study is applying Bayesian analysis in psychology and psychiatry, specifically suicidal study. This merit needs to be elaborated by providing the advantages of employing Bayesian inference in the field (e.g., reproducibility crisis). In fact, a previous study was done using a single-item measurement of suicidal ideation and the Bayesian analysis, which should be referred to as a typical example in this study: Nguyen MH, et al. (2021). Alice in Suicideland: Exploring the Suicidal Ideation Mechanism through the Sense of Connectedness and Help-Seeking Behaviors. International Journal of Environmental Research and Public Health, 18(7), 3681. https://doi.org/10.3390/ijerph18073681  

Third, one prominent advantage of using a single-item measurement lies in its capability to be analytically studied and incorporated with other theories and frameworks. For example, the above study used the Mindsponge Theory (Vuong, 2023) as its theoretical foundation, and the single-item measurement of suicidal ideation is highly compatible with the theory, making the study logically sound.

Vuong QH. (2023). Mindsponge Theory. De Gruyter. https://books.google.com/books?id=OSiGEAAAQBAJ

- Fourth, why does the study use only Item 1 of SBQ-R against Items 2-4, but not vice versa, like using Item 2 against Items 1, 3, 4, and so on? This shortage weakens the study's argument.

- Fifth, the main ideas of the study, either in the Introduction or Discussion, are ambiguous. Therefore, I suggest the authors split the Introdution and Discussion into multiple subsections and give each subsection a headline to increase the clarity of the study's main ideas.

Comments on the Quality of English Language

Minor editing of English language required

Reviewer 2 Report

Comments and Suggestions for Authors

This study used Bayesian network modeling and relative importance analyses on SBQ-R data from 1,160 U.S. and 1,141 Chinese undergraduate students. The Bayesian network analysis results showed that Item 1 is suitable for identifying other parameters of the suicide-related thoughts and behaviors construct. The results of the relative importance analysis further highlighted the relevancy of each SBQ-R item score when examining evidence for suicide-related thoughts and behaviors. These findings provided empirical support for using the SBQ-R item scores to understand the performances of different suicide-related behavior parameters. 

The introduction section is very lengthy. Please shorten it by highlighting the novelty of the current work.

How was the sample of size 1160 selected? What method is used for sample size determination? The authors should give details on this because this is the most important component of studies like this one. The same is true about the selection of Chinese students.

The complete computational code should be shared to verify the reproducibility.

Why only 10-fold crossvalidation is considered? Why not 5 or 15?

I also suggest the authors report MAPE.

As it is stated that 'Noteworthy is that our results are consistent with most of the findings in the extant literature'  then a question is what is new? The authors should report the insights.

Can the current work be extended by Meta-analysis? See, for example https://www.sciencedirect.com/science/article/pii/S0278584623000222

Reviewer 3 Report

Comments and Suggestions for Authors

The paper deals with very interesting topic of provided empirical support for using the SBQ-R item scores to understand 

the performances of different suicide-related behavior parameters. 

I found many methodological flaws in the paper:

1. What graphical model was used for the analysis. 

Was the polynomial model fitted in the bnlearn package as the default for discrete data?

2. Page 10, In Fig 1. This DAG was choesen form all data (separately for US or China) or from the crossvalidation?

If yes, please give the how often this model was chosen for repetition?

3. Page 11, Table 2. This conditional distribution was chosen by bootstap repetiotion?

These are averages from how many repetitions?

4. Page 13, Relative Importance Analysis within Regression.

Why the Authors examine the linear regression model for Items 2, 3, and 4 scores to predict scores on 

Item 1? This is a wrong idea because the direction of the arrows is the opposite (see Fig.1). 

The direction is from 1 to 2, 3, 4 , not reverse.

5. Page 13, why the Authors use the linear regresssion? Is is good for gaussian variables not for discreate variables.

The coefficient of determination R^2 for discrate variables is not proper. 

Minor remarks:

1. Pages 5-6, lines 254-272. It is good for gaussian variables not for discreate variables.

2. Page 6, line 285 states '10 transgender or not sure ...' but in Table 1 we have 'Transgender/ Not sure 19 '

Round 2

Reviewer 1 Report

Comments and Suggestions for Authors

All my comments have been addressed satisfactorily.

Comments on the Quality of English Language

Minor editing of English language required

Author Response

Thank you again for your detailed and constructive review. 

Reviewer 2 Report

Comments and Suggestions for Authors

This study used Bayesian network modeling and relative importance analyses on SBQ-R data from 1,160 U.S. and 1,141 Chinese undergraduate students. The Bayesian network analysis results showed that Item 1 is suitable for identifying other parameters of the suicide-related thoughts and behaviors construct. The results of the relative importance analysis further highlighted the relevancy of each SBQ-R item score when examining evidence for suicide-related thoughts and behaviors. These findings provided empirical support for using the SBQ-R item scores to understand the performances of different suicide-related behavior parameters. 

The authors addressed all my previous comments.

Author Response

(The authors gave the same response as above.)

Reviewer 3 Report

Comments and Suggestions for Authors

My previous comments were not taken into account. The work contains methodological errors. I can't accept this paper.

Author Response

Thank you again for taking the time to review this manuscript. The academic editor is following up with us on the comments.   

Round 3

Reviewer 3 Report

Comments and Suggestions for Authors

The Author response

"Using the SBQ-R, we treated the items (variables) as continuous, not dichotomous/discrete. We did not recode the items as dichotomous/discrete. Item 1 ranges from 1 to 4; Item 2 ranges from 1 to 5; Item 3 ranges from 1 to 3; Item 4 ranges from 0 to 6. A total score can thus be obtained by summing the scores on Items 1 to 4; the total score should range from 3 to 18."

I do not understand how variable ranges from 1 to 4 can be treated as continuous variable. Similar Item 3 from 1 to 3. I think that you should write that you used as a naive method Gaussian Graphical Model and compare it with results for more proper multinomal distribution.

"Examining predictors 2-4 for Item 1 is not inconsistent with the identified DAG, as the DAG represents conditional dependence relationships rather than directionality. " I agree that you can inference in oposite direction as in DAG but more natural is inference with the same direction as in  DAG.

Author Response

We thank for the constructive comments and suggestions and apologize for any confusion regarding the treatment of SBQ-R items in the manuscript. Please see the attachment.
